# Molecular-Targeted Fluorescence Lymph Node Imaging Could Play a Clinical Role in the Surgical Setting: A Systematic Review

**DOI:** 10.3390/cancers17081352

**Published:** 2025-04-17

**Authors:** Bo E. Zweedijk, Sebastiaan W. R. Dalmeijer, Labrinus van Manen, Hidde A. Galema, Lorraine J. Lauwerends, Hamed Abbasi, Bernd Kremer, Cornelis Verhoef, Dominic J. Robinson, Sjors A. Koppes, Alexander L. Vahrmeijer, Joost R. van der Vorst, Denise E. Hilling, Stijn Keereweer

**Affiliations:** 1Department of Otorhinolaryngology, Head and Neck Surgery, Erasmus MC Cancer Institute, University Hospital Rotterdam, Doctor Molewaterplein 40, 3015 GD Rotterdam, The Netherlandsh.abbasi@erasmusmc.nl (H.A.); d.robinson@erasmusmc.nl (D.J.R.); 2Department of Surgical Oncology and Gastrointestinal Surgery, Erasmus MC Cancer Institute, Doctor Molewaterplein 40, 3015 GD Rotterdam, The Netherlands; c.verhoef@erasmusmc.nl (C.V.);; 3Department of Surgery, Leiden University Medical Center, Albinusdreef 2, 2333 ZA Leiden, The Netherlands; 4Department of Otorhinolaryngology, Groene Hart Hospital, Bleulandweg 10, 2803 HG Gouda, The Netherlands; 5Department of Pathology, Erasmus Medical Center, Doctor Molewaterplein 40, 3015 GD Rotterdam, The Netherlands

**Keywords:** fluorescence imaging, fluorescence-guided surgery, image-guided surgery, molecular imaging, optical imaging, tumor-specific, intraoperative imaging, lymph node metastases

## Abstract

In many malignancies, the lymph node (LN) status is a critical prognostic factor and determines the need for (neo-) adjuvant therapies. Current diagnostic methods often miss smaller LN metastases (LNM). Over the past decade, intraoperative fluorescence imaging (FI) has emerged, with multiple studies evaluating molecular-targeted fluorescence tracers. As tumor-targeted fluorescent tracers have expanded, these procedures are increasingly used for LNM detection, raising hopes of addressing challenges with occult LNMs. This systematic review evaluates 26 studies on molecular-targeted FI for LNM detection. The results showed significant variability in sensitivity and specificity in the intraoperative setting, while postoperative FI on formalin-fixed tissue blocks demonstrated high sensitivity (70–100%) and specificity (66–100%) for LNM detection. Overall, molecular-targeted FI offers a limited clinical benefit for in vivo detection, but could improve the accuracy and efficiency of postoperative pathological processing.

## 1. Introduction

The lymphatic system plays a crucial role in the spread of solid tumors and is often the first site of metastasis, as cancer cells typically invade nearby lymph nodes (LN) before potentially spreading to other LNs through the lymphatic system and distant organs through the bloodstream [1]. In many malignancies, including colorectal, head and neck (HNC), melanoma, and breast cancer, the LN status is a vital prognostic factor for survival and determines the need for (neo-) adjuvant therapies [2,3,4]. Therefore, pre-, intraoperative, and postoperative nodal staging is essential to determine the optimal treatment and the need for (neo-) adjuvant therapy. Diagnostic imaging modalities (e.g., CT, MRI, PET-CT, SPECT-CT, and ultrasound-guided fine needle aspiration) are used to adequately stage the regional status of solid tumors, but smaller LN metastases (LNM) are still being missed in 10–30% of cases [5]. Fibroblast activation protein inhibitor (FAPI) PET/CT has recently emerged as a novel imaging technique, demonstrating superior target-to-background ratios compared to [18F] FDG PET/CT across a range of solid tumors. This higher contrast may enhance the sensitivity and specificity for detecting lymph node metastases, thereby improving staging accuracy and potentially informing patient management and therapeutic planning [6]. Additionally, intraoperative LN staging could be performed through sampling or a sentinel node (SN) procedure [7].

Over the past decade, a new era in the field of intraoperative imaging, using fluorescence imaging (FI), has emerged, as multiple clinical studies evaluating the use of molecular-targeted fluorescent tracers have been performed in recent years. Molecular-targeted fluorescent tracers, together with specialized camera systems, are essential tools in FI, that enable the precise intraoperative visualization of anatomical structures and pathological areas. These camera systems are able to detect the light emitted by the administered fluorescent tracer after excitation with a specific light source [8,9]. This provides the surgeon with real-time information on the location of the targeted cells. Applications of molecular-targeted FI have thus far mainly focused on the assessment of tumor resection margins and the detection of occult lesions. Non-targeted FI applications are more often used for the identification of vital structures (e.g., arteries, veins, and ureters) [10,11]. Two main application areas of molecular-targeted FI can be discerned, using either in vivo or ex vivo detection. In vivo assessment is defined as the utilization of FI in the patient during surgery. Conversely, ex vivo FI refers to the application of FI outside the patient to facilitate diagnostic evaluation and can be conducted either on the back-table during surgery, with the capacity to alter the surgical plan based on the imaging results, or postoperatively. With the growing variation of clinically available tumor-targeted fluorescent tracers, the scope of these intraoperative procedures has been increasingly expanded to include the detection of LNMs, raising expectations that it may provide a solution to the challenges associated with occult LNMs.

However, the different conjugates and fluorophores, the variety in NIR camera systems, and the differences in tissue optical properties require careful consideration in fluorescence-guided surgery [12]. NIR cameras differ in their sensitivity, resolution, and wavelength detection range, influencing the imaging efficacy. Additionally, tissue optical properties vary widely due to factors like composition, blood supply, and patient-specific characteristics (e.g., tissue vascularization, skin pigmentation), impacting light absorption, scattering, and emission. Consequently, ex vivo findings cannot be directly extrapolated to in vivo contexts.

This review is the first to summarize all clinical studies on molecular-targeted fluorescence imaging (FI) for detecting (occult) lymph node metastases (LNMs). Given the growing interest in novel FI tracers, there is a need for a structured framework to understand FI’s diverse applications in identifying LNMs, thus addressing the gaps in the current heterogeneous literature.

## 2. Materials and Methods

### 2.1. Literature Search and Study Selection

This systematic review was conducted according to the Preferred Reporting Items for Systematic Review and Meta-Analysis Protocols (PRISMA) guidelines [13]. The PRISMA checklist is included in Appendix B. We conducted a comprehensive literature search in four major databases (PubMed, MEDLINE, Embase, and Cochrane) to identify articles published prior to 12 January 2024 that discussed the use of molecular-targeted fluorescence LN imaging. Our search strategy utilized various terms and synonyms related to (near-infrared) fluorescence, optical imaging, molecular imaging, LNs with or without metastases, surgery, tumor, and targeted imaging to ensure a thorough review of relevant articles. Based on the screened articles, we grouped the tracers into the following four categories: antibody-based, folate receptor alpha, molecular-targeted peptides, and hybrid tracers. The complete search strategy is available in Appendix C. The article selection process involved three independent researchers (H.G., B.Z., and L.M.) who identified all relevant articles. Our review excluded articles that did not correlate imaging with histopathology. A Clinical Trial Number was not applicable.

### 2.2. Data Extraction

The general study characteristics, such as details on the fluorescent tracer, camera system, target tissue, study type, study population, number of resected LNs, intended clinical application, histopathology, definition of a positive fluorescent signal (e.g., tumor-to-background ratio, TBR), and fluorescence assessment (in vivo and ex vivo), were extracted. Furthermore, diagnostic test parameters (sensitivity, specificity, positive predictive value (PPV), and negative predictive value (NPV)) were extracted, if available, from the main text or Appendix A. If these parameters were not previously described, they were calculated using standard formulas based on the values of true-positives (TP), false-positives (FP), true-negatives (TN), and false-negatives (FN), when available. Sensitivity was calculated as TP/(TP + FN), specificity as TN/(TN + FP), PPV as TP/(TP + FP), and NPV as TN/(TN + FN).

True-positive lymph nodes (LNs) were defined as those that were fluorescent and contained metastases. False-negative LNs, on the other hand, were non-fluorescent, yet still harbored metastases. True-negative LNs were non-fluorescent and did not contain any metastases, while false-positive LNs were benign nodes that exhibited a fluorescent signal. If multiple doses or test characteristics were studied, the optimal dose/setting outcomes were included in this review.

### 2.3. Quality Assessment

Two reviewers (L.M. and B.Z.) independently evaluated the quality and risk of bias of the included articles using the revised Quality Assessment of Diagnostic Accuracy Studies tool [14]. An overview of this risk assessment is available in Appendix A. In case of discrepancies in interpretation, a third reviewer (S.K.) was consulted to adjudicate the decision.

## 3. Results

### 3.1. Overview of Included Studies

The literature search resulted in a total of 10,263 studies, as reported in a flow diagram (Figure 1). After title and abstract screening, a total of 263 articles underwent a full-text review. In total, twenty-six studies reported on molecular-targeted fluorescence LN imaging, and were therefore included in the final analysis (Table 1) [15,16,17,18,19,20,21,22,23,24,25,26,27,28,29,30,31,32,33,34,35,36,37,38,39,40]. The included studies were divided into different types of clinical trials (type A, B, C, or D) according to their methodology, using the classification introduced by Lauwerends et al. [41]. This classification is based on how and when fluorescence-guided surgery was introduced into the standard of care (Figure A1, Appendix A). The results of the quality and risk of bias assessment of the studies are provided in Figure A2 (Appendix A). Furthermore, we will describe both the in vivo and ex vivo assessments separately to elucidate their distinct advantages and applications.

### 3.2. In Vivo Assessment of Lymph Nodes

A total of fifteen studies discussed the in vivo application of molecular-targeted FI (Table 1). OTL38 contains a cyanine dye and targets the folate receptor alpha (FRα). Five studies evaluated the feasibility of in vivo LNMs detection using this dye in ovarian, endometrial, and gastric cancer, which also targets folate receptor alpha (FRα) [15,20,23,25,33]. Boogerd et al. showed that targeting FRα resulted in the detection of all 16 LNMs out of the 22 resected LNs in cases of endometrial cancer, with a mean in vivo TBR of 6.3 (SD-4.5; range 3.2–14.1). Even LNMs located under a tissue layer of ±1 cm could be visualized using the Quest Artemis camera system. Furthermore, a total of 50 benign LNs were resected in this study, of which 17 showed a fluorescent signal (mean TBR 2.5; SD = 1.3; range 1.5–6.2), resulting in a sensitivity, specificity, PPV, and NPV of 100%, 70%, 48%, and 100%, respectively [18]. Subsequent studies observed a high (up to 100%) rate of false-positive fluorescent lesions using OTL38, which were mainly benign LNs (based on binding of OTL38 to folate receptor β, vastly present in activated intranodal macrophages), showing that fluorescence-positive lesions should be interpreted with caution [15,23,25,33].

The in vivo targeting of the folate receptor alpha (FRα) has also been reported in several other studies, of which Tummers et al. described the feasibility for detection of primary breast and ovarian cancer using EC17 (emission 520 nm), including the aim to detect LNMs in vivo. In this study, they found only one fluorescent LN, which turned out to be false-positive; the total of resected LNs was not described [16].

[111In] In-DOTA-labetuzumab-IRDye800CW is a dual-labeled anti-carcinoembryonic antigen (anti-CEA) antibody conjugate. De Gooyer et al. demonstrated that deeper-seeded suspect lesions (e.g., retroperitoneal LNs) that were identified on SPECT imaging were not detectable with in vivo FI using [111In] this dye [36]; in all, 17 out of 28 (61%) malignant lesions could be detected with NIR-fluorescence imaging in the 2 mg group. In the 10 mg group, 95% (n = 16) of all malignant lesions could be visualized, and in the 50 mg group, all 25 (100%) malignant lesions could be detected. Additionally, an undetected retroperitoneal LNM was both visualized using SPECT-CT and FI prior to cytoreduction in one patient, and FI revealed four previously undetected LNMs, resulting in a change in treatment plan in two patients [36]. In both studies, no data on the fluorescence intensity of the LNs (such as Mean Fluorescence Intensity (MFI) or TBR) were provided, and the authors did not include information on the sensitivity, specificity, PPV, or NPV.

Additionally, in vivo targeting of the FRα has been reported in several studies, of which Tummers et al. described the feasibility of the detection of primary breast and ovarian cancer using EC17, including the aim to detect LNMs in vivo.

In this study, they found one fluorescent LN, which turned out to be false-positive; the total of resected LNs was not described. Three studies explored in vivo LN imaging during colorectal cancer surgery using a fluorescently labelled monoclonal antibody targeting the carcinoembryonic antigen receptor (SGM-101). Further assessment of the LNs was conducted ex vivo [21,31,40]. In a subsequent pilot study, one fluorescent retroperitoneal-located LNM was detected in vivo, in addition to ten false-positive LNs. Information concerning the false-positive and false-negative fluorescent signals of the resected LNs was lacking in all of these studies [21,31,40].

During robotic prostate cancer surgery using IS-002 (prostate-specific membrane antigen (PSMA)-targeted peptide-based fluorescent tracer), eight out of fourteen LNMs were detected in vivo, resulting in a sensitivity of 57.1% [38]. Similar results in prostate cancer patients were reported with another PSMA-targeted tracer, OTL78, which could detect four out of seven conglomerate LNMs in vivo, also resulting in a sensitivity of 57.1% [39]. Both studies showed a high number of false-positive LNs (81 out of 309, 26%) [38] and conglomerates of LNs (13 out of 59, 22%)) [39], leading to a relatively low PPV of 73.8% and 55.2%, respectively.

In a study on colorectal cancer, EMI-137, a c-MET-targeted, fluorescently labeled tracer was evaluated. No detectable fluorescent LN signal was observed in vivo in any participant (n = 9). Additionally, the systematic administration of EMI-137 and the fluorescent assessment using the Karl Storz^®^ laparoscopic system did not facilitate the detection of LNMs in this trial [35].

Further analysis of fifteen malignant nodes via immunohistochemistry revealed moderate to high c-MET expression in all samples. However, the c-MET expression did not correlate with the fluorescence signal observed on microscopy slides. Consequently, it remains challenging to draw definitive conclusions about the efficacy of EMI-137 in detecting LNMs through FI [35]. Two studies investigated the use of Bevacizumab (targeting the vascular endothelial growth factor α, VEGF-α) bound to either IRDye800CW or 800CW in the identification of primary breast cancer and locally advanced rectal cancer, respectively. As a secondary objective, the in vivo detection of LNMs was evaluated. A fluorescent signal was detected in the LNM of only one out of five patients with LNMs during breast cancer surgery.

Moreover, the fluorescent signal was evaluated alongside immunohistochemically staining in 104 LN tissue blocks, demonstrating no difference in VEGF-α expression between benign and LNMs [17].

### 3.3. Ex Vivo Assessment of Lymph Nodes

Ex vivo assessment of LNs can be performed either intraoperatively or postoperatively. Ex vivo assessment during the surgical procedure allows for immediate clinical decision-making, potentially aiding the surgeon and directly benefiting the patient. Postoperative assessment, on the other hand, primarily assists the pathologist, contributing to the overall efficiency, efficacy, logistics, and cost-effectiveness of the process (see Figure 2). In this context, we will describe both approaches separately to elucidate their distinct advantages and applications. Eight studies examined the ex vivo use of molecular-targeted fluorescence imaging (FI) during surgery, while six studies focused on its ex vivo application postoperatively. Additionally, two studies covered both intraoperative and postoperative ex vivo applications (see Table 1).

#### 3.3.1. Ex Vivo Assessment During the Surgical Procedure

Cetuximab-IRDye800CW and Panitumumab-IRDye800CW are fluorescent tracers that both target the endothelial growth factor receptor (EGFR) in HNC. The ex vivo identification of LNMs using Cetuximab-IRDye800CW demonstrated high sensitivity (97.2%) and specificity (92.7%). Furthermore, out of 471 resected LNs, 69 LNs were fluorescent, of which 35 were true-positive, and 34 were false-positive, resulting in a positive predictive value of 50.7%. Only one LNM was missed in these twelve patients (one false-negative) [18].

In a colorectal cancer study, the CEA-targeting agent SGM-101 was used to identify colorectal cancer. Among thirty-seven patients who underwent surgery, eleven LNs were resected due to clinical suspicion, and metastases were confirmed in six of these nodes. Out of the six LNMs, five were detected ex vivo using FI, with a relatively low TBR ranging from 1.4 to 1.9 (true-positives). One LNM was non-fluorescent (false-negative), and details regarding the remaining resected LNs were not provided in the study [31]. SGM-101 was also utilized in thirteen patients with colorectal cancer and lung metastases.

Among these, two benign LNs were resected on white light suspicion in two patients, but they were fluorescence-negative (true-negatives). In another patient a lymphadenectomy was performed for preoperatively identified hilar LNMs, and three malignant LNs were fluorescent on the back-table (true-positives). Furthermore, three other non-fluorescent LNs were resected based on clinical suspicion for tumor involvement, and all three contained fibrosis without a tumor (true-negatives). Further details were not provided in the study report [40].

In a study by Jonker et al., EMI-137 was evaluated for detecting LNMs using FI in papillary thyroid cancer (PTC). The fluorescence signal was assessed in 76 malignant and 340 benign lymph nodes (LNs), both ex vivo on fresh specimens and after pathological processing. If one or more lymph nodes within a level had a fluorescence intensity above the device-specific threshold, this was defined as a fluorescence-positive level. The tests showed an LN level-specific sensitivity (sensitivity specified per anatomical level stage) of 87.5%, a specificity of 26.3%, and an NPV of 83.3% for detecting levels containing PTC-positive LNs with micro-metastatic foci at gross examination. The authors concluded that if these LNs would be assessed ex vivo during surgery, this could have additional value for the surgeon to rule out the presence of nodal metastases in the central compartment, improving the selection of patients that may benefit from omitting a prophylactic central LN dissection, thus leading to a reduction in overtreatment and associated morbidity in the management of PTC [37].

In pancreatic cancer, three studies assessed the use of FI to detect LNMs ex vivo during surgery. Two studies demonstrated that panitumumab-IRDye800CW and ex vivo FI could distinguish tumor-bearing LNs from negative ones based on a statistically significant difference in MFI [22,29]. In the first of these two studies, FI successfully identified LNMs. During surgery, tumor-bearing LNs could be identified with a mean TBR of 6.3 ± 0.82. Furthermore, tumor-bearing LNs (n = 29) could be detected with significantly higher MFI (0.06 ± 0.01) compared to tumor-negative (n = 78) LNs (0.02 ± 0.002) (*p* < 0.001) [22].

In the second study, cetuximab-IRDye800 was tested in two dose cohorts (50 mg vs. 100 mg) across 144 lymph nodes (LNs) originating from seven patients. In the low-dose group, 72 LNs were evaluated, with 89% being tumor-negative and 11% being tumor-positive; in the high-dose group, 78% were tumor-negative and 22% were tumor-positive. Overall, tumor-positive LNs showed a significantly higher mean fluorescence intensity (MFI) (0.06 vs. 0.02; *p* < 0.001). This difference was more pronounced in the low-dose cohort (0.07 vs. 0.02; *p* < 0.001), while in the high-dose group, the MFI difference was not significant. The low-dose cohort showed high sensitivity and specificity, with a likelihood ratio of 4.6 for detecting tumor-positive LNs. In the high-dose cohort, both the sensitivity and specificity decreased, with a likelihood ratio of 1.3. Furthermore, of the seventeen LNs with an “occult” tumor (defined as tumor foci < 5 mm in size), fifteen were detected by fluorescence, and two were not, resulting in a sensitivity of 88%. The amount of tumors in the two non-fluorescent LNs was comparable to the fluorescent LNs (0.83 mm vs. 1.98 mm, *p* = 0.425), and the NIR FI enhanced the visualization of LNMs, and even small (≤ 2 mm) peritoneal metastases [29].

In addition, ex vivo FI performed during surgery, as well as the postoperative and pathologic assessment of LNMs, showed a sensitivity between 88% and 100%, and a specificity between 32% and 78%, in favor of the low-dose [26].

In breast cancer patients undergoing SN or axillary LN dissection, a ratiometric activatable peptide-based tracer (AVB-620) was evaluated ex vivo. Ratiometric measurements involve assessing AVB-620 uptake by comparing the fluorescence intensity in tumor tissue relative to adjacent non-tumor tissue, enabling a consistent and quantitative comparison of tracer accumulation. AVB-620 was administered either the day before surgery (12–20 h prior) or on the day of surgery (2–12 h prior). With day-of-surgery dosing, the average ex vivo TBR was 1.09 ± 0.18 compared to 0.59 ± 0.04 for adjacent tissue. It is of note that a minimum TBR of 1.5 is generally accepted to discriminate fluorescent lesions [42]. For day-prior dosing, the signal-to-background ratio (SBR) was 1.90 ± 0.18 compared to 1.17 ± 0.20 for adjacent tissue. Identified tumor regions of interest (ROIs) corresponded with surgeon-identified disease and pathology reports. Although the authors noted that positive LNs had a higher ratio than negative nodes, the number of positive nodes, and thus the detection efficacy outcome parameters, were not reported [19].

In locally advanced rectal cancer, patients received an intravenous bolus injection of 4.5 mg of bevacizumab-800CW, a fluorescent tracer targeting VEGF-α, 2–3 days before surgery. Ex vivo FI of the LNs showed a fluorescence signal in two enlarged LNs, which proved to be tumor-positive. However, this study did not provide further details on the total number of imaged LNs and the number of true-/false-positive/negative LNs [27].

Furthermore, two studies evaluated the feasibility of a hybrid fluorescent and radioactive targeted tracer in metastasized colorectal cancer undergoing cytoreductive surgery ([111In] In-DOTA-labetuzumab- IRDye800CW) and prostate cancer surgery ([18F]-BF3- Cy3-ACUPA) [30,36]. Aras et al. describe that, in one patient, four pelvic LNs were resected based on PET-imaging, and the ex vivo back-table FI showed that two LNs had a higher intensity and were confirmed to be LNMs. No further data about the resected LNs were described [30]. De Gooyer et al. demonstrated that the tracer is effective for sensitive imaging at doses of 10 or 50 mg, with a back-table TBR of 2.62 (SD 0.49) and 2.73 (SD 0.94), respectively. Preoperative imaging detected previously unidentified LNM in one patient, while intraoperative FI revealed additional LNMs in two patients. This multimodal imaging approach led to changes in the clinical strategy for three patients, supporting its potential to enhance surgical decision-making in cytoreductive procedures. Details on false-positive and false-negative results were not provided [36].

#### 3.3.2. Ex Vivo Assessment After the Surgical Procedure

Postoperative assessment of LNMs using FI could improve the timely and efficient assessment of LN dissection specimens by the pathologist. Furthermore, preoperative administration of fluorescent tracers for post-surgical pathology assessment may seem unconventional. Given the promising results from the growing number of studies using FI to detect primary tumor margins both in vivo and ex vivo, it could be expected that a significant number of patients will receive tumor-targeted tracers in the future. For many solid tumors, FI could serve as a potential tool in surgery for assessing primary tumor margins and aiding the postoperative evaluation of LNs during pathological assessment.

If a FI threshold—a specific fluorescent intensity value used to differentiate between benign and malignant LNs—could be found that secures a 100% NPV, the amount of LNs that need pathological assessment could potentially be reduced significantly, making pathology assessment less time-consuming and more efficient [26]. In addition, while direct visualization and palpation are currently considered the gold standard for pathological assessment in detecting and resecting LNs, this method is labor-intensive, can overlook smaller LNs, and is subject to high interobserver variability, potentially leading to missed occult metastases [43].

In a study of oral cancer, 960 LNs were analyzed ex vivo, of which 34 (3.5%) were found to contain metastatic disease. Panitumumab-IRDye800CW demonstrated preferential localization to LNMs, as indicated by a higher fluorescent signal in these nodes compared to other LNs. The median MFI of LNMs was significantly higher than that of benign LNs (0.06 versus 0.02, *p* < 0.05) [32]. In addition, a selection of five LNs with the highest fluorescence intensity from individual specimens yielded a sensitivity of 100%, a specificity of 85.8%, and an NPV of 100% for detecting occult metastases, resulting in 100% accuracy in clinical neck staging within this study.

In the clinically node-positive (cN+) cohort, evaluating a selection of five LNs with the highest fluorescence intensity per patient demonstrated a sensitivity of 87.5%, a specificity of 93.2%, and an NPV of 99.1% for detecting LNMs [32]. Figure 3 provides an overview of both in and ex vivo fluorescence imaging of cervical lymph nodes, including the corresponding Hematoxylin and Eosin (H&E) slides. Although many benign lymph nodes exhibit fluorescence (Figure 3B), ranking the nodes based on mean fluorescence intensity reveals that metastatic lymph nodes display the highest fluorescence (Figure 3D), suggesting sufficient contrast for differentiation.

Five studies performed postoperative FI of formalin-fixed paraffin-embedded (FFPE) tissue blocks on a microscopic level [24,26,29,34,39]. Fluorescence-positivity was determined using variable definitions (thresholding the mean fluorescence intensity [34]); maximum fluorescence intensity [34]; TBR [29]; or combined mean FI and TBR [24,26,39]). In case the TBR was used, the background was defined as either adjacent tissue [26,29,39] or fibro-adipose tissue [24]. This procedure was detailed by Nishio et al. in a study involving 24 patients (1012 LNs analyzed) undergoing surgery for head and neck cancer following the systemic injection of Panitumumab-IRDYe800CW [24]. FI allowed for the identification of LNMs with a high sensitivity (85%) and specificity (94%), which could potentially reduce the number of LNs undergoing pathological processing and examination by 90%. These results were in concordance with another study, using Cetuximab-IRDye800CW, in 22 HNC patients (514 LNs analyzed), that demonstrated ex vivo detection of LNMs with a 100% sensitivity, 87% specificity, 49% PPV, and a 100% NPV using optimal mean FI thresholding. The authors concluded that this could serve as a selection tool for pathologists by reducing the amount of LNs necessitating microscopic examination by 77.4%, without overlooking LNMs. Importantly, in 7.5% of initially identified as false-positive LNs, tumor deposits were subsequently detected, underscoring instances missed by conventional histopathological analysis [34].

In a colorectal cancer study, the feasibility of identifying LNMs ex vivo was investigated using an integrin-targeted fluorescent tracer (cRGD-ZW800-1). A total of 209 LNs were excised, of which 35 contained metastases. Ex vivo FI demonstrated a sensitivity and NPV of 100%, with a specificity and PPV of 87% and 33%, respectively. Since no false-negative LNs were reported, these findings indicate that the pathologist could selectively process only the fluorescent LNs, potentially reducing the workload of pathological examination by approximately 50% [28]. In addition, for patients with pancreatic cancer, the use of Cetuximab-IRDYe800CW demonstrated a significant difference in ex vivo mean fluorescence intensity (MFI) between tumor-positive and tumor-negative LNs (*p* < 0.001, with a dose-dependency of *p* = 0.03). The study’s microscopic detection of LNMs revealed high sensitivity (91–93%) but low to moderate specificity (35–66%).

Therefore, it is crucial that FI can even detect LNMs with minimal amounts of tumor, enhancing postoperative pathology analysis. Of the seventeen LNMs with an “occult” tumor (tumor foci < 5 mm in size), fifteen were detected by fluorescence, and two were not, resulting in a sensitivity of 88% [24]. In line with Cetuximab-IRDYe800CW, Panitumumab-IRDYe800CW showed a significantly higher TBR and MFI in LNMs, compared to benign LNs, in patients with pancreatic cancer.

However, they found a relatively lower sensitivity (68%) and higher specificity (92%). Of the 67 LNMs, 19 were not detected using FI, resulting in an NPV of 93% [29]. More recently, Stibbe et al. evaluated the use of OTL78 during prostate cancer surgery in 18 patients. Post hoc ex vivo imaging identified four (57%) of seven metastatic LN conglomerates. There were 21 false-positive LN clusters. OTL78 dose de-escalation and interval prolongation led to a lower false-positive LN cluster rate and a higher false-negative rate, in vivo and ex vivo. Post hoc gross-macroscopic and microscopic fluorescence imaging of individual LNs (n = 289) identified all nine LNMs with a median SBR of 3.5 (IQR 2.7–5.3) for gross-macroscopy and 8.8 (IQR 3.1–20.3) for microscopy. Because only tumor-positive and/or fluorescent microscopic slides were selected, requested, and imaged, specificity and false-positive rates were not calculated in this study [39].

## 4. Discussion

The development of molecular-targeted FI for improved intraoperative visualization of primary tumors has led to significant interest in its potential for detecting LNMs. This article presents the first systematic review of clinical trials evaluating the use of molecular-targeted FI for perioperative in vivo and ex vivo detection of LNMs.

Clinical evaluations of molecular-targeted FI include both intraoperative and postoperative LN imaging. In the intraoperative setting, both in vivo and ex vivo FI techniques were employed. The use of fluorescent imaging and its diagnostic value requires more thorough reporting, as the diagnostic accuracy of in vivo molecular-targeted FI for detecting LNMs was reported in only four of the twenty-six studies reviewed [20,31,38,39].

Correspondingly, studies evaluating the detection of LNs report significant variability in sensitivity, specificity, PPV, and NPV. In some cases, data on LNs and LNMs were not reported, which underscores the lack of clear reporting. It is important to have all data available to correctly interpret the results. Furthermore, while most studies demonstrate high TBRs for LNMs, numerous false-positive LNs were also observed. For example, in benign LNs, the panitumumab-IRDye800CW signal appears to result from non-specific accumulation, demonstrated by a dose-dependent increase in fluorescent antibody-complex detection within lymphatic channels, despite the absence of EGFR expression [24]. Additionally, false-negative LNs were also noted, potentially due to several factors, such as the following: the absence of the targeting molecule in LNMs, the limited penetration depth of the NIR light for LNs covered by tissues thicker than 1 cm, or the limited camera sensitivity in detecting minimal fluorescence in small (micro-)metastases.

Importantly, ex vivo postoperative molecular-targeted FI on FFPE tissue blocks demonstrated both high sensitivity and specificity in all studies for detecting LNMs. High sensitivity is particularly clinically important for confidently ruling out the presence of metastases in LNs requiring pathological assessment. Additionally, intraoperative FI for detecting LNMs was primarily performed ex vivo on resected specimens, with fluorescence signals later correlated with histopathological findings, and less frequently conducted in vivo. Since most studies were exploratory, the diagnostic accuracy of FI for LN detection was reported in approximately one-third of the included studies, with only one study providing both in vivo and ex vivo test characteristics, which underlines the problem of reporting and extrapolating ex vivo data to in vivo purposes [39].

The lack of detailed reporting on critical factors in scientific papers underscores the limited recognition of their importance, making our systematic review of this topic both crucial and highly relevant. The key issue driving this systematic review is the significant difference in imaging conditions between in vivo and ex vivo settings, which prevents the direct extrapolation of ex vivo findings to in vivo applications. Variations in tissue optical properties—such as composition, blood supply, vascularization, and skin pigmentation—combined with the limited penetration of NIR light, can greatly affect light absorption, scattering, and emission, influencing LN visualization in vivo [12]. Furthermore, other factors, such as the device sensitivity, the dose, and the pharmacological properties of the fluorescent agent, could be influential [6]. In contrast, ex vivo imaging allows for easier tissue manipulation and more standardized conditions. Recognizing these differences is essential for accurately interpreting the study results and evaluating LN visualization in vivo, highlighting the necessity of our comprehensive review.

Ultimately, intraoperative FI should serve as a tool for surgical decision-making in vivo, which requires reports on how the detection of LNMs could assist during surgical procedures. Our review indicates that FI using tumor-targeted tracers is not yet feasible in detecting (occult) LNMs in vivo, and thus the technique of optical imaging, because of its intrinsic limitations, may also not be the most suitable approach in this context. However, it does have an impact on the postoperative workflow, as it can reduce the workload of the pathologist by, for instance, prioritizing the assessment of higher-fluorescent LNs instead of all LNs, thereby creating a more efficient workflow. Only one study documented clinical decision-making based on the correlation of in vivo fluorescence with intraoperative assessment by the pathologist [31]. While the sensitivity for detecting LNMs was 80% and the specificity was 60%, these findings did not lead to changes in the surgical plan. Several clinical implications for using intraoperative FI for LN detection, both in vivo and ex vivo, can be proposed, but the current systems seem insufficient for detecting “occult” LNMs in vivo.

The current limitations of FI technology prevent it from replacing the SN procedure. While fluorescence imaging is a valuable tool for aiding in the detection of SNs, it is not yet capable of fully replacing the conventional approach due to the challenges associated with NIR camera systems. Light absorption and scattering in tissue reduce the depth at which fluorescence signals can be detected, limiting the visualization of deeper nodes. Additionally, variations in tissue properties (e.g., vascularization, pigmentation, variability in tissue types) can affect the fluorescence signal, leading to inconsistent results. Furthermore, the current fluorescence technique cannot distinguish between the SN and other LNs; typically, the first fluorescent lymph node is interpreted as the SN. These factors prevent the state-of-the-art FI from matching the accuracy and reliability of the SN procedure, currently making it a complementary rather than standalone method for intraoperative SN detection.

Theoretically, FLI could influence intraoperative decision-making in several ways, which are as follows:

Firstly, in vivo detection of occult LNMs could necessitate the abortion of the surgical procedure. For example, in colorectal cancer, detecting peritoneal lesions and LNs suspicious for metastases using FI and confirming them with fresh frozen sectioning could be critical in intraoperative decision-making, potentially leading to an open-and-close procedure. Such a finding would be a significant game-changer, as it directly impacts the surgical approach and patient management. However, while FI should serve as a tool for identifying LNMs, its state-of-the-art technology is not powerful enough to rule out occult LNMs [41].

Secondly, real-time detection of LNMs could assist in guiding the extent of lymphadenectomy, potentially reducing the scope of surgery and associated morbidity. For instance, during thyroid surgery, FI could potentially reduce the number of resected benign LNs by more than 25%. However, its diagnostic accuracy needs to be evaluated in vivo in future studies, as this particular study was only conducted ex vivo [18,32,37].

Lastly, another possible application of in vivo LN imaging could be to improve the tumor margin assessment in case of extra nodular extension of LNMs, which is associated with worse survival. FI could potentially help visualize margin delineation and could therefore be clinically relevant when the ingrowth extends beyond the nodal capsule [44].

A possible solution to the limitations of FI for preoperative and intraoperative imaging could be optoacoustic imaging (OAI), which may improve imaging depth by several centimeters compared to the less than 1 cm depth achieved with FI. The principles of OAI are similar to those of FI, but OAI detects acoustic waves generated by the optical excitation of the fluorophore, rather than using NIR wavelengths. Therefore, combining FI and OAI could be beneficial for the non-invasive detection of LNMs. However, small tumor deposits in occult LNMs make detection with both FI and OAI challenging due to the reduced excitation of the fluorophore in these small deposits. This was demonstrated in a proof-of-concept study by Vonk et al., where they explored the use of multispectral optoacoustic tomography (MSOT) for the in vivo detection of LN metastases in oral cancer patients. The study highlighted the potential of MSOT in improving preoperative detection, as it allows for the visualization of both intrinsic tissue chromophores and administered contrast agents at depths of several centimeters, making it suitable for identifying early-stage metastases, though limitations related to signal intensity and fluorophore concentration were observed [45,46].

This review highlights a potential role for FI in the postoperative assessment of LNs. Pathological LNMs could be successfully distinguished from benign LNs through fluorescence assessment, as LNs can be preselected based on their MFI and TBR with high specificity. This could influence the LN assessment workflow, reducing the number of LNs that require pathological processing and histological examination. Furthermore, the high sensitivity suggests that analyzing a smaller number of fluorescent LNs could be more efficient and less labor-intensive, leading to better detection of LNMs, more accurate postoperative staging and treatment, and fewer missed LNMs compared to standard histopathologic analysis.

The studies included here reported various definitions for fluorescence-positive LNs, including subjective assessments by surgeons and more quantitative measures using thresholds (e.g., TBR, MFI, or a combination of both). Although studies have demonstrated that it is possible to distinguish between benign and malignant LNs using currently available semi-quantitative methods, the lack of consensus on standard fluorescence assessment restricts a fair comparison between clinical studies [47]. Therefore, future advancements in methods and techniques that support the accurate quantification of fluorophore concentrations in tissue, for instance, fluorescence lifetime imaging, could enhance the objective differentiation between tumor and non-tumor tissue [11,48].

## 5. Conclusions

While molecular-targeted fluorescence imaging (FI) has demonstrated the ability to detect LNMs using tumor-specific tracers, it has not yet shown in vivo benefits for the detection of LNMs, particularly in the context of occult metastases. Potential applications for intraoperative LN imaging have been described and suggested. However, FI’s effectiveness as a tool, combined with or without other techniques, and its impact on surgical planning, are infrequently reported. Nevertheless, the postoperative assessment of LNs using molecular-targeted FI has shown high accuracy, indicating its potential to distinguish pathologically confirmed LNMs from benign ones. This could potentially increase efficiency and reduce the need for extensive pathological processing and histological examination. Further research is warranted to fully elucidate the clinical utility of FI in the in vivo setting.

## Figures and Tables

**Figure 1 cancers-17-01352-f001:**
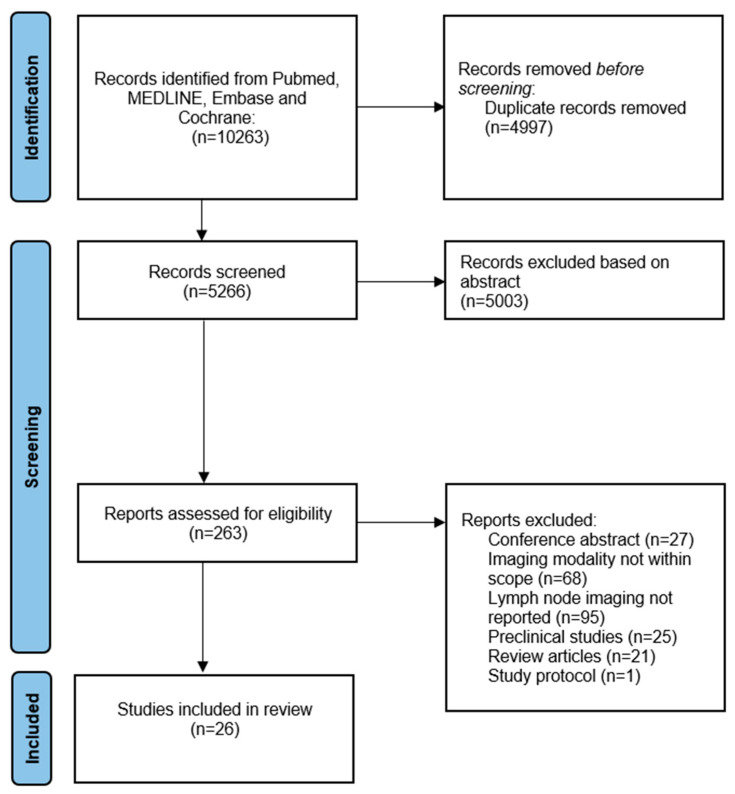
Flow chart of study inclusions.

**Figure 2 cancers-17-01352-f002:**
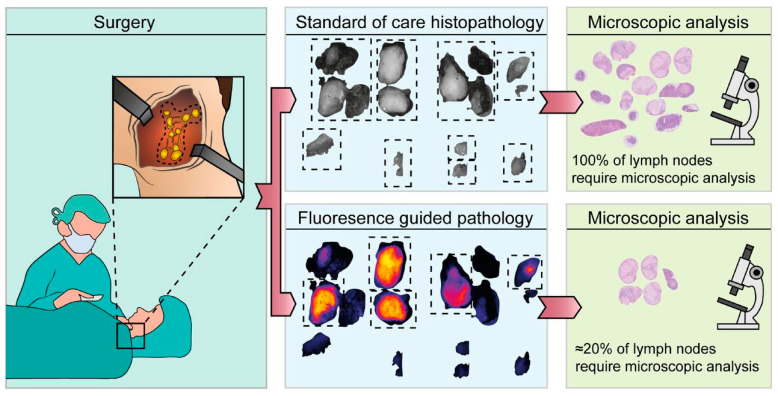
Application of fluorescence molecular imaging postoperatively. This research was originally published in JNM by Vonk et al. Epidermal Growth Factor Receptor–Targeted Fluorescence Molecular Imaging for Postoperative Lymph Node Assessment in Patients with Oral Cancer. Journal of Nuclear Medicine May 2022, 63 (5) 672–678; DOI: https://doi.org/10.2967/jnumed.121.262530©SNMMI [34].

**Figure 3 cancers-17-01352-f003:**
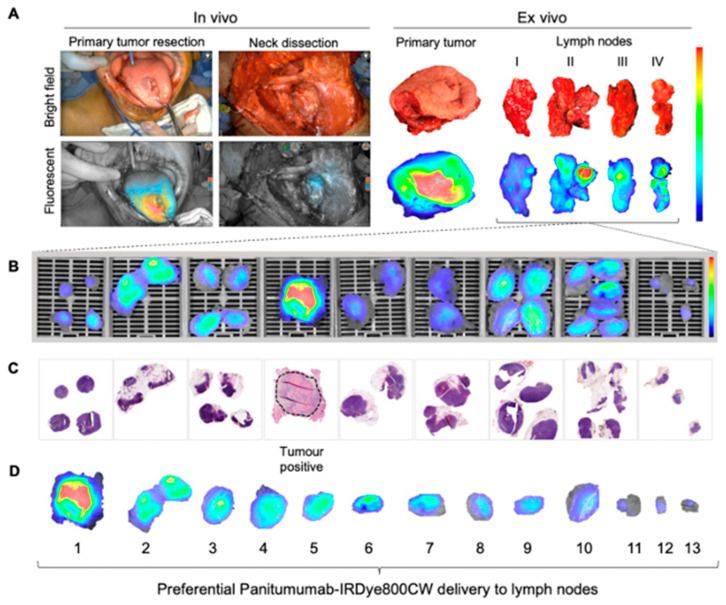
Fluorescence imaging of cervical lymph nodes: (**A**) Intraoperative bright field and fluorescence imaging (in and ex vivo). (**B**) Postoperative ex vivo fluorescence imaging of formalin-fixed lymph node slides. (**C**) Corresponding H&E slides showing tumor status, with the most fluorescent LN being tumor-positive (dotted circle). (**D**) Ranking individual LNs by mean fluorescence intensity. This research was originally published in Theranostics by Krishnan et al. Metastatic and Sentinel Lymph Node Mapping Using Intravenously Delivered Panitumumab-IRDye800CW. Theranostics 2021;11:7188–7198 [32].

**Table 1 cancers-17-01352-t001:** Overview on included studies.

Study	Reference	Target	Tracer	Target Tissue	Clinical Trial Design	Patients (n)	Harvested Lymph Nodes (n)	Used Fluorescence Camera System(s)	Intended Clinical Application	Sensitivity (%)	Specificity (%)	PPV (%)	NPV (%)
**Boogerd et al., 2018**	[21]	Carcinoembryonic antigen	SGM-101	Colorectal cancer	Type E	26	N.S.	- Quest Artemis and Spectrum imaging system - PEARL imaging system	Intraoperative tumor and metastases detection				
**De Valk et al., 2021**	[31]	Carcinoembryonic antigen	SGM-101	Colorectal cancer	Type E	37	11 (all doses)	Quest Spectrum Imager	Intraoperative tumor and metastases detection (focusing on change in treatment strategy)	83 * (in vivo/ex vivo)	60 * (in vivo/ex vivo)	71 * (in vivo/ex vivo)	75 * (in vivo/ex vivo)
**De Gooyer et al., 2022**	[36]	Carcinoembryonic antigen	[111In]In-DOTA-labetuzumab- IRDye800CW	Metastasized colorectal cancer	Type E	15	N.S.	Quest Spectrum Imager	Intraoperative tumor and metastases detection				
**Meijer et al., 2023**	[40]	Carcinoembryonic antigen	SGM-101	Colorectal lung metastases	Type E	13	8	- Quest Spectrum Imager - PEARL imaging system	Intraoperative metastases detection				
**Armstrong et al., 2022**	[35]	c-MET	EMI-137	Colon cancer	Type C	9	15	Karl Storz laparoscopic camera system	Intraoperative tumor and metastases detection				
**Jonker et al., 2022**	[37]	c-MET	EMI-137	Papillary thyroid cancer	Type C	19	289	IVIS spectrum imaging	Intraoperative lymph node metastases detection	88 ** (ex vivo)	26 ** (ex vivo)	N.S. ** (ex vivo)	83 ** (ex vivo)
**Rosenthal et al., 2017**	[18]	Epidermal growth factor receptor	Cetuximab-IRDYe800CW	Head-and neck cancer	Type C	12	471	- Luna Imaging System - PEARL imaging system	Intraoperative lymph node metastases detection	97 (ex vivo)	93 (ex vivo)	51 (ex vivo)	100 (ex vivo)
**Tummers et al., 2018**	[22]	Epidermal growth factor receptor	Cetuximab-IRDye800	Pancreatic ductal adenocarcinoma	Type E	7	107	- Laparoscopic PINPOINT 9000 system - Wide-field SurgVision Explorer - PEARL imaging system	Intraoperative tumor and lymph node metastases detection				
**Nishio et al., 2019**	[24]	Epidermal growth factor receptor	Panitumumab-IRDye800CW	Head-and neck cancer	Type C	24	1012	- Spy-Phi camera and IR9000 optical imaging platform - PEARL imaging system	Postoperative lymph node metastases detection before pathological examination	94 (FFPE)	85 (FFPE)	36 (FFPE)	99 (FFPE)
**Tummers et al., 2019**	[26]	Epidermal growth factor receptor	Cetuximab-IRDYe800CW	Pancreatic cancer	Type C	7	72 (low-dose cohort)	- Wide-field SurgVision Explorer - PEARL imaging system	Intraoperative lymph node metastases detection	100 (ex vivo) 91 (FFPE)	78 (ex vivo) 66 (FFPE)	36 * (ex vivo) N.S. (FFPE)	100 * (ex vivo) N.S. (FFPE)
**Lu et al., 2020**	[29]	Epidermal growth factor receptor	Panitumumab-IRDYe800CW	Pancreatic cancer	Type C	11	132	- Laparoscopic PINPOINT 9000 system - Spy-Phi camera system - Wide-field SurgVision Explorer - PEARL imaging system - IGP-ELVIS imaging system	Intraoperative tumor and metastases detection	70 (FFPE)	91 * (FFPE)	62 (FFPE)	93 (FFPE)
**Krishnan et al., 2021**	[32]	Epidermal growth factor receptor	Panitumumab-IRDYe800CW	Head-and neck cancer	Type C	27	581 (cN0 cohort)	- Spy-Phi camera and IR9000 optical imaging platform - PEARL imaging system	Intraoperative (sentinel) lymph node metastases detection	100 (ex vivo)	86 (ex vivo)	100 * (ex vivo)	100 (ex vivo)
**Vonk et al., 2021**	[34]	Epidermal growth factor receptor	Cetuximab-IRDYe800CW	Oral cancer	Type B	22	514	PEARL imaging system	Postoperative lymph node metastases detection before pathological examination	100 (FFPE)	87 (FFPE)	49 (FFPE)	100 (FFPE)
**Tummers et al., 2016**	[16]	Folate receptor	EC17	Ovarian cancer and breast cancer	Type E	15	N.S.	Quest Artemis imaging system	Intraoperative tumor and metastases detection				
**Hoogstins et al., 2016**	[15]	Folate receptor	OTL38	Ovarian cancer	Type E	12	13	Quest Artemis imaging system	Intraoperative tumor and metastases detection				
**Boogerd et al., 2018**	[20]	Folate receptor	OTL38	Endometrial cancer	Type E	4	66	Quest Artemis imaging system	Intraoperative tumor and metastases detection	100 (in vivo)	70 (in vivo)	48 (in vivo)	100 (in vivo)
**Hoogstins et al., 2019**	[23]	Folate receptor	OTL38	Ovarian cancer	Type E	6	38	Quest Artemis imaging system	Intraoperative metastases detection				
**Randall et al., 2019**	[25]	Folate receptor	OTL38	Ovarian cancer	Type E	48	N.S.	- Quest Artemis imaging system - Novadaq PINPOINT LI system - Visionsense VS3 imaging system	Intraoperative tumor and metastases detection				
**Newton et al., 2021**	[33]	Folate receptor	OTL38	Gastric cancer	Type E	5	N.S.	Visionsense VS3 Iridium imaging system	Intraoperative tumor and metastases detection				
**De Valk et al., 2020**	[28]	Integrins (associated with tumor angiogenesis)	cRGD-ZW800-1	Colon cancer	Type B	12	209	- Olympus Visera Elite I - Quest Spectrum Imager - PEARL imaging system	Intraoperative tumor and metastases detection	100 (ex vivo)	87 (ex vivo)	33 (ex vivo)	100 (ex vivo)
**Unkart et al., 2017**	[19]	Matrix metalloproteinases	AVB-620	Breast cancer	Type C	26	N.S.	N.S.	Intraoperative tumor and lymph node metastases detection				
**Aras et al., 2021**	[30]	Prostate-specific membrane antigen	[18 F]-BF3- Cy3-ACUPA	Prostate cancer	Type B	10	N.S.	Custom made	Intraoperative tumor and metastases detection				
**Stibbe et al., 2023**	[39]	Prostate-specific membrane antigen	OTL78	Prostate cancer	Type E	18	20 (optimal dose cohort)	- Da Vinci Si or Xi Surgical System - VisionSense near-infrared imaging system	Intraoperative tumor and metastases detection	0 (in vivo, ex vivo) 100 (FFPE) 100 (microscopy)	100 (in vivo, ex vivo) 100 (FFPE) N.S. (microscopy)	N.S. (in vivo, ex vivo) 100 (FFPE) 100 (microscopy)	90 (in vivo, ex vivo) 100 (FFPE) N.S. (microscopy)
**Nguyen et al., 2023**	[38]	Prostate-specific membrane antigen	IS-002	Prostate cancer	Type D	24	309 (all doses)	- Da Vinci Si or Xi Surgical System	Intraoperative tumor and metastases detection	57 * (in vivo)	73 * (in vivo)	9 (in vivo)	97 (in vivo)
**Lamberts et al., 2017**	[17]	Vascular endothelial growth factor	Bevacizumab-IRDye800CW	Breast cancer	Type E	20	N.S.	Custom made (Technical University Munich)	Intraoperative tumor and metastases detection				
**De Jongh, et al., 2020**	[27]	Vascular endothelial growth factor	Bevacizumab-800CW	Locally advanced rectal cancer	Type B	17	N.S.	Wide-field SurgVision Explorer	Intraoperative tumor (focusing on margins) and metastases detection				

* Test characteristics were calculated based on the provided prevalence and true-/false-positives and true-/false-negatives. ** Level-specific test characteristics were provided; no diagnostic accuracy was provided on lymph node detection. Abbreviations: FFPE: formalin-fixed, paraffin-embedded; NPV: negative predictive value; N.S.: not specified, PPV: positive predictive value.

## Data Availability

The data that support the findings of this study are available on request from the corresponding author.

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
