# Peer review of "Molecular-Targeted Fluorescence Lymph Node Imaging Could Play a Clinical Role in the Surgical Setting: A Systematic Review"

_cancers, 2025, doi:10.3390/cancers17081352_

Round 1
Reviewer 1 Report
Comments and Suggestions for Authors
Congratulations to the authors for the work presented. I found the issue related to the use of FI in finding LMNs lymph node metastases very interesting. Interest that is reflected in surgical practice and clinical practice for the different types of therapeutic approach, clinical risks and advantages. Understanding the right use and the most sensitive and specific method is the point of arrival.
Well done the research study and the approach. Good writing and scientific references.
I ask to go a little beyond the results and in the discussion express some ideas on the specific difficulties in the different organ-specific studies. What can be the limitations of lymph node uptake? (the dose, type of organ and structure, devices, pharmacological contrast, functional clinical conditions?)
Thanks and good work
Author Response
Congratulations to the authors for the work presented. I found the issue related to the use of FI in finding LMNs lymph node metastases very interesting. Interest that is reflected in surgical practice and clinical practice for the different types of therapeutic approach, clinical risks and advantages. Understanding the right use and the most sensitive and specific method is the point of arrival.
Well done the research study and the approach. Good writing and scientific references.
I ask to go a little beyond the results and in the discussion express some ideas on the specific difficulties in the different organ-specific studies. What can be the limitations of lymph node uptake? (the dose, type of organ and structure, devices, pharmacological contrast, functional clinical conditions?)
Thanks and good work
Dear reviewer 1, we would like to thank you for your valuable comments and are appreciative of your time reviewing our manuscript. We have made significant changes throughout, to comply with your comments.
Comment 1: I ask to go a little beyond the results and in the discussion express some ideas on the specific difficulties in the different organ-specific studies. What can be the limitations of lymph node uptake? (the dose, type of organ and structure, devices, pharmacological contrast, functional clinical conditions?)
Response: Thank you for this insightful comment. In response, we have expanded the discussion to address the potential limitations of lymph node uptake across different organ-specific studies. Please see the revised paragraph beginning at line 484, where we elaborate on the influence of factors such as dosage, anatomical and structural differences between organs, imaging devices, the type of fluorescent contrast agents used, and patient-specific clinical conditions. Additionally, we have included a new sentence at line 490 to further highlight these considerations.
Reviewer 2 Report
Comments and Suggestions for Authors
As per as the systematic reviews are concerned, the authors needs to clearly indicate the below points very minutely, but it is not mentioned in the paper clearly.
1.Overview of lymph node metastasis mechanisms in cancer
2. Comparison of molecular imaging techniques for detecting lymph node metastasis
3. Recent advancements in nanobiotechnology and how significantly it improved targeted drug delivery systems aimed at addressing lymph node metastasis in cancer.
4. Integration of imaging and therapeutics
5. Distinguishing sentinel lymph nodes from regional lymph nodes in the context of cancer treatment.
Moreover challenges/ Obstacles and future outlooks/perspectives in molecular imaging and targeted therapies for lymph node metastasis in cancer needs to be mentioned very minutely in the aspect of diagonistics methods, clinical trials, regulatory approvals,targetted therapeutics which is not mentioned properly.
The diagrams are very basic outline given, authors are required to design more constructive designs in respect to the technical concerns are there.
The author may add some nanobiotechnolical advances as well in the targetted drug delivery for lymph node metatasis in cancer.
Moreover only 48 referrences are there. A review must include more than minimum 250-300 reff to share these study in a explanatory way.
Author Response
As per as the systematic reviews are concerned, the authors needs to clearly indicate the below points very minutely, but it is not mentioned in the paper clearly.
1.Overview of lymph node metastasis mechanisms in cancer
- Comparison of molecular imaging techniques for detecting lymph node metastasis
- Recent advancements in nanobiotechnology and how significantly it improved targeted drug delivery systems aimed at addressing lymph node metastasis in cancer.
- Integration of imaging and therapeutics
- Distinguishing sentinel lymph nodes from regional lymph nodes in the context of cancer treatment.
Moreover challenges/ Obstacles and future outlooks/perspectives in molecular imaging and targeted therapies for lymph node metastasis in cancer needs to be mentioned very minutely in the aspect of diagonistics methods, clinical trials, regulatory approvals,targetted therapeutics which is not mentioned properly.
The diagrams are very basic outline given, authors are required to design more constructive designs in respect to the technical concerns are there.
The author may add some nanobiotechnolical advances as well in the targetted drug delivery for lymph node metatasis in cancer.
Moreover only 48 referrences are there. A review must include more than minimum 250-300 reff to share these study in a explanatory way.
Dear reviewer 2, we would like to express our gratitude towards your time spend to review our manuscript and for your valuable comments. We have made significant improvements to the manuscript, based on your comments, which are described below.
Comment 1: Overview of lymph node metastasis mechanisms in cancer
Response: Thank you for this suggestion. We have addressed this point by adding content in sentences 57 and 58 to provide a brief overview of the mechanisms involved in lymph node metastasis. Additionally, we have included a reference supporting the mechanisms of lymph node metastasis in cancer.
Comment 2: Comparison of molecular imaging techniques for detecting lymph node metastasis.
Response: We appreciate the comment. This point is already addressed in the introduction, specifically in sentence 62.
“Diagnostic imaging modalities (e.g., CT, MRI, PET-CT, SPECT-CT, and ultrasound-guided fine needle aspiration) are used to adequately stage the regional status of solid tumors, but smaller LN metastases (LNM) are still being missed in 10-30% of cases. [5]”
Comment 3: Recent advancements in nanobiotechnology and how significantly it improved targeted drug delivery systems aimed at addressing lymph node metastasis in cancer.
Response: Thank you for this insightful comment. The focus of this systematic review was on the diagnostic applications of molecular-targeted fluorescence imaging, rather than on therapeutic strategies or drug delivery systems. We have added a brief description of the mechanisms underlying lymph node metastasis, revised the introduction accordingly, and included modalities such as FAPI to highlight recent advancements in addressing lymph node metastasis in cancer. Furthermore, we would like to emphasize that comprehensive reviews on the use of fluorescence-guided surgery remain scarce in the current literature.
Comment 4: Integration of imaging and therapeutics
Response: We appreciate the interest in this area; however, our review specifically focuses on the diagnostic value of molecular fluorescence imaging for lymph node metastases. Therapeutic applications were beyond the intended scope of this study. Please also refer to our response to Comment 3.
Comment 5: Distinguishing sentinel lymph nodes from regional lymph nodes in the context of cancer treatment
Response: Thank you for raising this important point. We have addressed this issue in the paragraph beginning at sentence 508, where we discuss the current limitations of fluorescence techniques in distinguishing between sentinel lymph nodes and other lymph nodes. Furthermore, we have added clarification in sentence 513 and included an additional reference (Reference 7: Baldari L, Boni L, Cassinotti E. Lymph node mapping with ICG near-infrared fluorescence imaging: technique and results. Minim Invasive Ther Allied Technol. 2023;32:213–221. doi:10.1080/13645706.2023.2217916).
Comment 6: Moreover challenges/ Obstacles and future outlooks/perspectives in molecular imaging and targeted therapies for lymph node metastasis in cancer needs to be mentioned very minutely in the aspect of diagonistics methods, clinical trials, regulatory approvals,targetted therapeutics which is not mentioned properly.
Response: We appreciate your comment. While our review is primarily focused on diagnostic imaging, we have included a discussion of the current challenges and limitations of fluorescence-guided surgery (FGS) in the paragraph beginning at sentence 484. These limitations stem from technical, biological, and clinical implementation aspects. Although some studies suggest that FGS alone could offer a solution, we believe that due to its inherent limitations, a multimodal approach combining techniques such as SPECT and PET may be necessary for optimal lymph node detection and staging. The scope of this review is to provide an overview of FGS studies, rather than to focus on targeted therapies.
Comment 7: The diagrams are very basic outline given, authors are required to design more constructive designs in respect to the technical concerns are there
Response: Thank you for the suggestion. We have added Figure A2, titled "Light propagation through tissue," illustrating how light traveling through tissue is subject to reflection, scattering, and absorption. This figure is adapted from: Stijn Keereweer, Pieter B.A.A. Van Driel, Thomas J.A. Snoeks, Jeroen D.F. Kerrebijn, Robert J. Baatenburg de Jong, Alexander L. Vahrmeijer, Henricus J.C.M. Sterenborg, Clemens W.G.M. Löwik; Optical Image-Guided Cancer Surgery: Challenges and Limitations. Clin Cancer Res 15 July 2013; 19(14): 3745–3754. https://doi.org/10.1158/1078-0432.CCR-12-3598. This addition provides further insight into the technical challenges associated with the technique.
Comment 8: The author may add some nanobiotechnolical advances as well in the targetted drug delivery for lymph node metatasis in cancer
Response: Thank you for this insightful comment. The focus of this systematic review was on the diagnostic applications of molecular-targeted fluorescence imaging for lymph node metastases, rather than on therapeutic strategies or drug delivery systems. We have added a brief description of the mechanisms underlying lymph node metastasis, revised the introduction accordingly, and included modalities such as FAPI to highlight recent advancements in addressing lymph node metastasis in cancer. While we appreciate the interest in therapeutic applications, these were beyond the intended scope of this review. Furthermore, we would like to emphasize that comprehensive reviews on the use of fluorescence-guided surgery remain scarce in the current literature. Please also refer to our response to Comment 3 and 4 for additional clarification.
Comment 9: Moreover only 48 references are there. A review must include more than minimum 250-300 reff to share these study in an explanatory way.
Response: We acknowledge the relatively limited number of references. However, since this systematic review targets a specific and emerging diagnostic technique, the current body of literature is still limited. We have included all relevant and eligible studies that met our predefined criteria, in accordance with PRISMA guidelines.